# Early-warning signals for Dansgaard-Oeschger events in a high-resolution ice core record

Niklas Boers [1,2]

The Dansgaard–Oeschger (DO) events, as observed in oxygen isotope ratios from the North Greenland Ice Core Project (NGRIP) record, are an outstanding example of past abrupt climate transitions. Their physical cause remains debated, and previous research indicated that they are not preceded by classical early-warning signals (EWS). Subsequent research hypothesized that the DO events are caused by bifurcations of physical mechanisms operating at decadal timescales, and proposed to search for EWS in the high-frequency fluctuation levels. Here, a time series with 5-year resolution is obtained from the raw NGRIP record, and significant numbers of EWS in terms of variance and autocorrelation increases are revealed in the decadal-scale variability. Wavelet analysis indicates that the EWS are most pronounced in the 10–50-year periodicity band, confirming the above hypothesis. The DO events are hence neither directly noise-induced nor purely externally forced, which provides valuable constraints regarding potential physical causes.

---

[1] Grantham Institute for Climate Change, Imperial College, London SW7 2AZ, UK. [2] Potsdam Institute for Climate Impact Research, Potsdam 14473, Germany. Correspondence and requests for materials should be addressed to N.B. (email: nboers@ic.ac.uk)

The last glacial interval, which lasted roughly from 110 to 12 kyears before 2000 AD (b2k), experienced abrupt climatic transitions at millennial timescales, which are called Dansgaard–Oeschger (DO) events[1–3]. They manifest themselves most concisely in time series of oxygen isotope ratios ($\delta^{18}$O) and dust concentrations retrieved from Greenland ice core records[3,4], yet their impact is noticeable in different archives across the globe. The DO events are characterized by rapid, decadal-scale transitions of $\delta^{18}$O from Greenland stadial (GS) to Greenland interstadial (GI) conditions, which are followed by comparably slow relaxations back to GS conditions within centuries or millennia. Oxygen isotope ratios in ice cores provide a qualitative proxy of the atmospheric temperature at the location of the ice core, and temperature increases of up to 16.5 °C have been reported for single DO events[5].

An apparent periodicity for DO events at 1470 years has been suggested[6] and put into context with astronomical forcing by centennial-scale solar cycles[7]. However, it has also been shown that the waiting times between subsequent DO events are within the high-likelihood range of an exponential distribution, implying that the timing of the DO events is consistent with a Poisson process. The proposed periodicity could, therefore, in fact be spurious[8].

Although different theories to explain the DO events have been proposed, there is no agreement concerning the responsible physical mechanisms. A key question to constrain the set of potential explanations is whether the DO transitions are induced by external forcing, by noise, or by bifurcations in the underlying dynamical system. In the latter case, the DO events might be preceded by statistical early-warning signals (EWS), such as increases in variance and autocorrelation[9–11]. Heuristically, these two EWS can be associated with a widening of the basin of attraction prior to the transition, and for certain classes of low-dimensional random dynamical systems, they can be inferred rigorously[12], e.g. in analogy with the Ornstein–Uhlenbeck process[11]. It should be emphasized, however, that when studying complex natural systems with many degrees of freedom, the absence of statistical EWS in the corresponding time series does not necessarily imply that the transitions are not caused by bifurcations: Different factors, such as interactions with other relevant variables, potentially across different timescales, may hide the EWS even if the transition is caused by a simple fold bifurcation. Moreover, the bifurcation structure itself may not be simple enough to cause simple EWS in terms of variance and autocorrelation increases.

If statistical EWS are observed, it is typically assumed that the corresponding transitions are not entirely noise-induced: If a transition is not caused by a bifurcation, but by increasing noise level of the underlying random dynamical system, the variance of the time series increases, but the autocorrelation remains constant. On the other hand, if the transition is caused, e.g. by a fold bifurcation, both variance and autocorrelation are expected to increase.

The fact that a forthcoming bifurcation implies the presence of EWS does, however, not exclude the possibility that statistical fluctuations indistinguishable from EWS arise either by chance or by other mechanisms unrelated to a bifurcation or abrupt transition. In other words, the presence of statistical fluctuations resembling EWS does not necessarily imply that a bifurcation is approaching, because these fluctuations might be caused by other reasons as well. Such false positives would reduce the predictive skill of a forecast scheme based on EWS. Typically, EWS are only searched for in the parts of a time series that precede abrupt transitions, which have themselves been identified beforehand. Even in this case, statistical fluctuations that could be mistaken for EWS may arise by chance. This issue, which is related to multiple comparisons, calls for a statistical test regarding the number of observed EWS, in addition to testing the significance of each single EWS.

It was previously shown that the $\delta^{18}$O record obtained from the North Greenland Ice Core Project (NGRIP[4]), when interpolated to a time series with 50-year temporal resolution, does not exhibit EWS if the DO events are considered individually, and if the entire frequency spectrum is taken into account[11]. On the other hand, weak EWS have been reported when the ensemble average of DO events 2–16 (in the stratigraphic counting from younger to older events) is considered[13]. Such an ensemble average of EWS may, however, be strongly dominated by single events, and a cross-validation to exclude this possibility has not been performed in the latter study.

Recently, it was demonstrated that significant EWS can in fact be observed at least for some of the DO events individually, if the analysis is restricted to the high-frequency variability of the NGRIP $\delta^{18}$O time series[14]. The latter study is based on the more recent 20-year mean time series of the NGRIP record[15], for the time period from 60 to 11 kyears, and focusses on the periodicity band between 40 and 60 years. Instead of the total variance and autocorrelation, scale-averaged wavelet coefficients ($\hat{w}^2$) and local Hurst exponents ($\hat{H}^{\mathrm{loc}}$), respectively, are considered. In this context, $\hat{w}^2$ is used to estimate the variance of high-frequency fluctuations, and $\hat{H}^{\mathrm{loc}}$ is interpreted as an estimator of the correlation time in the high frequencies. The latter provides a suitable replacement of the autocorrelation, since an increasing $\hat{H}^{\mathrm{loc}}$ would signal an increase in correlation time, consistent with a destabilization of the attractor. The physical hypothesis behind this approach is that there exists a subcomponent of the underlying dynamical system that operates at decadal timescales, which is responsible for the decadal-scale DO transitions via a bifurcation. According to this hypothesis, EWS would be detectible in the isolated high-frequency band but masked otherwise by mechanisms operating at longer timescales. For the ensemble average of the 17 DO transitions during the studied interval, Rypdal[14] found a significant ($p < 0.05$) increase of the variance of the high-frequency fluctuations. However, since the corresponding ensemble average of the autocorrelation was not studied, the significant increase of the variance could be explained by an increase of the noise level alone, instead of an approaching bifurcation. Furthermore, these ensemble results are probably dominated by the transition to the Bølling–Allerød interstadial (i.e. DO-1): When considered individually, only two DO events (DO-1 and DO-8) are preceded by wavelet coefficient increases that are significant with $p < 0.05$, and only one (DO-1) is preceded by a significant increase of the Hurst exponent[14]. These numbers of EWS could, in fact, arise purely by chance.

Here the promising hypothesis of Rypdal[14] is re-evaluated with some technical modifications, using the recently published raw data of the NGRIP record, which allow for much higher temporal resolution. The raw NGRIP $\delta^{18}$O and dust values are measured at 5-cm steps in the ice core[4], and using the associated, annual layer-counted chronology[16,17], these values can be interpolated to time series with regular 5-year sampling steps (Fig. 1). Using these $\delta^{18}$O and dust time series, it is thus possible to study periodicities of the NGRIP record at timescales as low as 10 years.

We refer to the Methods section below for details on the data preprocessing, as well as for definitions for the scale-averaged wavelet coefficient $\hat{w}^2$ and local Hurst exponent $\hat{H}^{\mathrm{loc}}$, which are used to estimate the variance and correlation times, respectively, in the high-frequency bands of the $\delta^{18}$O and dust time series.

EWS are first searched for in terms of significant positive trends in the variance ($\hat{\sigma}^2(t)$) and lag-1 autocorrelation ($\alpha_1(t)$) of the 100-year high-pass filtered NGRIP data. Specifically, we apply a Chebyshev Type-I high-pass filter with cutoff at 100 years to the time series and then estimate $\hat{\sigma}^2(t)$ and $\alpha_1(t)$ in sliding windows of width equal to 200 years. Thereafter, the corresponding wavelet-based estimators $\hat{w}^2$ and $\hat{H}^{\mathrm{loc}}$, which are also averaged in time windows of 200 years for consistency, are analysed with particular

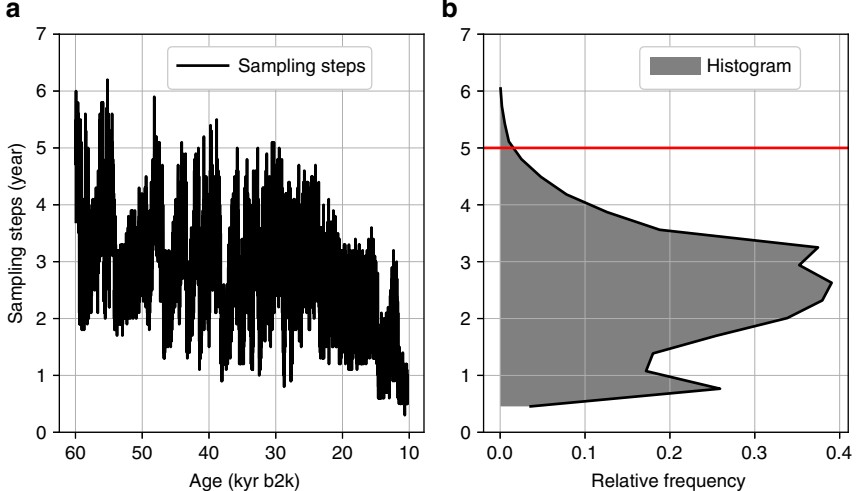

**Fig. 1** Temporal sampling steps of the Greenland Ice Core Chronology 2005. **a** Temporal sampling steps of the Greenland Ice Core Chronology 2005 (GICC05), as a function of time. Note how the time intervals between subsequent measurement increase with age. **b** Histogram of the temporal sampling steps. The horizontal red line marks the temporal resolution of 5 years, to which the raw data are interpolated. Only 0.4% of the temporal sampling steps are >5 years

focus on identifying the specific frequency ranges that carry the most pronounced EWS.

## Results

**Early-warning signals in the variance and autocorrelation.** For each abrupt transition, we first search for significant EWS in the preceding GS period, until 200 years prior to the transition to ensure that no data from the transition itself is taken into account. Note that the high-resolution version of NGRIP record extends only to 59,939 years b2k, and we thus do not have sufficient data to investigate whether DO-17, which occurred approximately at 59,450 years b2k, is preceded by EWS. We thus investigate 16 DO events, as well as the Younger-Dryas/Preboreal transition at approximately 11,705 years b2k.

The statistical significance of trends in the different EWS estimators is determined on the basis of surrogates with randomized phases, with the null hypotheses that the trend is equal to zero (Methods). Throughout this paper, an EWS is called significant if the corresponding $p$ value is <0.05.

As noted above, statistical fluctuations that resemble EWS may arise by chance, potentially leading to false positives. To account for this issue, which is related to the general problem of multiple comparisons, the following significance test for the total number of observed EWS is introduced: We first choose 17 time points (corresponding to the observed number of abrupt transitions) randomly from the respective time series. Thereafter, we count how many of these are preceded by significant increases of the EWS estimator in question, within time spans for which the duration is chosen in accordance with the original GS intervals. Repeating this 10,000 times, a null model distribution is obtained for the number of EWS preceding random time points. This significance test is constructed separately for EWS in the $\sigma^2$ time series and the $\alpha_1$ time series, as well as for EWS in both of them at the same time. Corresponding tests are constructed for the wavelet-based estimators $\hat{w}^2$ and $\hat{H}^{\text{loc}}$ (Supplementary Figure 1).

When searching for EWS during the GS intervals of the 100-year high-pass filtered $\delta^{18}O$ time series, significant EWS are found for 12 out of the 17 events in the variance $\sigma^2(t)$. In the lag-1 autocorrelation $\alpha_1(t)$, significant EWS are found for 7 out of the 17 events, and for 6 out of the 17 events, EWS are detected in both $\sigma^2(t)$ and $\alpha_1(t)$ simultaneously (Fig. 2b, c). According to the test described above, these numbers are statistically significant at 95% confidence level (Table 1 and Supplementary Figure 1).

It seems natural to restrict the search for EWS to the GS intervals prior to the transitions[14]. Recall, however, that the hypotheses is that a dynamical subsystem operating at decadal timescales causes the transitions via a bifurcation and that the imprints of this bifurcation are masked by mechanisms with longer (centennial to millennial) timescales. If this hypotheses is correct, then the fast subsystem might, after the bifurcation and the resulting transition, return to its prior state much faster than suggested by the slow relaxation from GI to GS conditions within centuries to millennia. In such a situation, one may argue that EWS should, as in ref. [13], be searched for in the entire intervals between subsequent transitions and not only during GS intervals.

Indeed, when searching for EWS in the entire time interval between subsequent transitions, with an offset of 200 years at the beginning and at the end of the interval, similar numbers of EWS are detected: We find 13 significant EWS for $\sigma^2(t)$ and 7 significant EWS in $\alpha_1(t)$, as well as in both of them simultaneously (Fig. 2e, f) These numbers of EWS are statistically significant (Table 1 and Supplementary Figure 1).

It might be criticized that this number of significant EWS, obtained when analysing the entire intervals between subsequent transitions, is simply due to the fact that two different physical regimes—the GI and GS intervals, respectively—are considered: it is true that the high-frequency variance $\sigma^2(t)$ is generally lower during GI than during GS intervals. However, this is not the case for the autocorrelation coefficient $\alpha_1(t)$ and, moreover, the physical cause of the lower $\sigma^2(t)$ during GI intervals may be directly related to the physical cause of the observed statistical EWS themselves. Ultimately, it will only become possible to answer this question once an unambiguous physical theory of the DO events is established, because that would allow to explain the statistical EWS in terms of physical processes. Since such a theory has not yet been obtained, the results for both options are presented here.

**Wavelet-based early-warning signals.** As an alternative to $\sigma^2(t)$ and $\alpha_1(t)$, it was suggested by Rypdal[14] to consider the scale-averaged wavelet coefficient $\hat{w}^2(t)$ and the local Hurst exponent $\hat{H}^{\text{loc}}(t)$ of the $\delta^{18}O$ time series. When restricting the latter two estimators to the 10–50 year periodicity band, we observe significant correlations with $\sigma^2(t)$ and $\alpha_1(t)$: $\rho(\sigma^2(t), \hat{w}^2(t)) = 0.97$, and $\rho(\alpha_1(t), \hat{H}^{\text{loc}}(t)) = 0.71$. This indicates that the wavelet-based estimators are indeed suitable as EWS indicators.

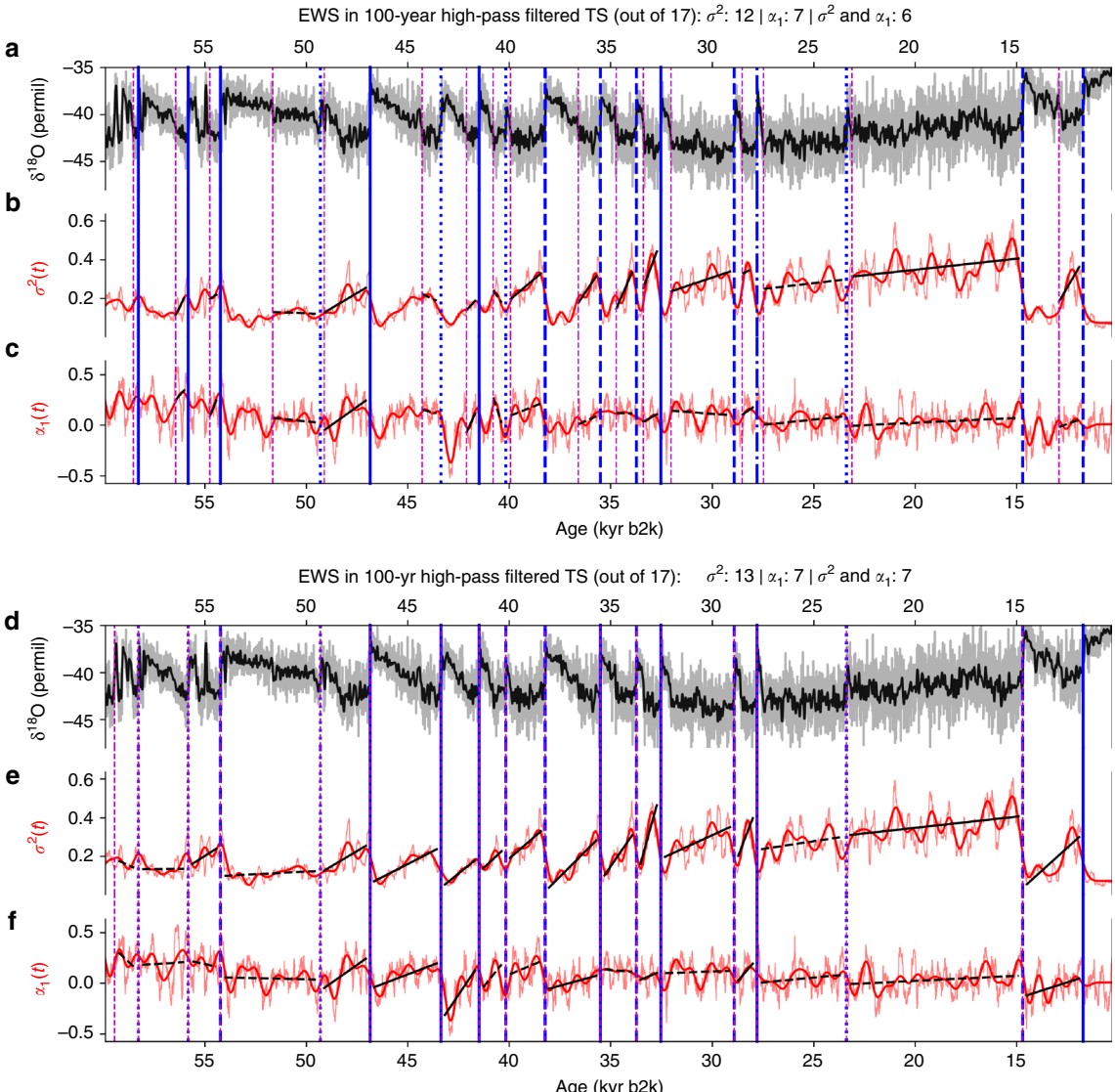

**Fig. 2** Early-warning signals in the variance and autocorrelation. **a** The 5-year-interpolated $\delta^{18}O$ time series (grey), together with a low-pass filtered version (black) for visual clarity. Dansgaard–Oeschger (DO) events, as well as the Younger-Dryas/Preboreal transition, are marked by the vertical blue lines, and the preceding ends of GI intervals by vertical dashed magenta lines (cf. Supplementary Table 1 for corresponding ages). EWS are searched for in the GS intervals between the magenta and blue lines. **b** Time series of the variance $\sigma^2$ of the 100-year high-pass filtered $\delta^{18}O$ time series (red, see text for details). **c** Same as **b**, but for the lag-1 autocorrelation coefficient $\alpha_1$. **d–f** Same as **a–c**, but here, EWS are searched for in the entire intervals between subsequent DO transitions. Significant (non-significant) trends according to the phase-randomization test (see Methods) are indicated by the solid (dashed) black lines. Transition events that are preceded by significant EWS in $\sigma^2$ ($\alpha_1$) are marked by dashed (dash-dotted) blue lines, and transitions that are preceded by EWS in both $\sigma^2$ and $\alpha_1$ are indicated by solid blue lines. If neither of the two estimators shows significant EWS, the blue lines are dotted

When restricting the search for EWS to the GS intervals and the 10–50 year periodicity band, we find 12 significant EWS in $\hat{w}^2(t)$, 8 significant EWS in $\hat{H}^{loc}(t)$ and 7 significant EWS simultaneously in both (Fig. 3b, c). Again, these numbers are significant at 95% confidence according to the test described above (Table 1 and Supplementary Figure 1). While we find 13 EWS in $\hat{w}^2(t)$ when searching for EWS in the entire intervals between DO transitions, the number of EWS in $\hat{H}^{loc}(t)$ and hence also the number of simultaneous EWS in both is reduced to 3 (Fig. 3e, f). Although a comparably small number, this is still a statistically significant result (Table 1 and Supplementary Figure 1). Note that, if the Morlet wavelet basis is used in this situation instead of the Paul wavelet basis, 14 significant EWS are found in $\hat{w}^2(t)$, and 6 significant EWS in $\hat{H}^{loc}(t)$, as well as in both estimators simultaneously (Supplementary Figure 2).

Remarkably, the corresponding analysis of the dust time series reveals a substantially smaller (and not significant) number of EWS (Table 1 and Supplementary Figure 3), although the dust time series itself is quite strongly correlated with the $\delta^{18}O$ time series (the Pearson correlation is $\rho = 0.76$).

The fact that we find significant numbers of simultaneous EWS in $\sigma^2(t)$ and $\alpha_1(t)$ of the 100-year high-pass filtered $\delta^{18}O$ time series, as well as in $\hat{w}^2$ and $\hat{H}^{loc}$ confined to the 10–50 year periodicity band of the same variable, provides direct evidence for a widening of the basin of attraction associated with the hypothesized mechanism operating at decadal scales. To find the periodicity band with the highest number of significant EWS in the $\delta^{18}O$ time series, the above analysis of the wavelet-based estimators is repeated for different values of the lower ($s_1$) and upper ($s_2$) bound of the investigated periodicity band, namely, all combinations with $s_2 \in$

**Table 1 Numbers of observed significant EWS**

|  |  | GS | | GI and GS | |
|---|---|---|---|---|---|
|  |  | $\delta^{18}O$ | Dust | $\delta^{18}O$ | Dust |
| $\sigma^2(t)$ and $\alpha_1(t)$ |  |  |  |  |  |
| $\sigma^2$ | 95% confidence | 9 | 9 | 7 | 7 |
|  | No. of EWS | **12** | 2 | **13** | 3 |
| $\alpha_1$ | 95% confidence | 7 | 7 | 5 | 5 |
|  | No. of EWS | 7 | 7 | **7** | 3 |
| $\sigma^2$ and $\alpha_1$ | 95% confidence | 5 | 5 | 3 | 3 |
|  | No. of EWS | **6** | 1 | **7** | 1 |
| $\hat{w}^2(t)$ and $\hat{H}^{loc}(t)$ |  |  |  |  |  |
| $\hat{w}^2$ | 95% confidence | 9 | 9 | 7 | 7 |
|  | No. of EWS | **12** | 5 | **13** | 3 |
| $\hat{H}^{loc}$ | 95% confidence | 7 | 7 | 4 | 4 |
|  | No. of EWS | **8** | 6 | 3 | 1 |
| $\hat{w}^2$ and $\hat{H}^{loc}$ | 95% confidence | 4 | 4 | 2 | 2 |
|  | No. of EWS | **7** | 4 | **3** | 0 |

Numbers of EWS are shown for the variance $\sigma^2(t)$ and lag-1 autocorrelation $\alpha_1(t)$ of the 100-year high-pass filtered NGRIP $\delta^{18}O$ and dust time series, as well as for the scale-averaged wavelet coefficient $\hat{w}^2(t)$ and Hurst exponent $\hat{H}^{loc}(t)$, confined to the 10–50 year periodicity bands of the NGRIP $\delta^{18}O$ and dust time series. Significance thresholds for the numbers of EWS at 95% confidence are presented for comparison (see text for details on the significance test). Numbers of EWS are reported not only for the situation where EWS are searched for in Greenland stadials (GS) prior to the transitions but also for the situation where EWS are searched for within the entire interval between subsequent transitions (GS and GI). Numbers above their respective significance thresholds are shown in bold

[20, 30,…, 100, 110], and $s_1 \in [10,…, s_2 - 10]$ such that $s_1 < s_2$. When searching for EWS only in the GS intervals prior to the transitions, $\hat{w}^2$ exhibits a maximum of 12 EWS for several combinations of $s_1$ and $s_2$, but with $s_1$ at most equal to 30 years (Fig. 4a). For $\hat{H}^{loc}$, less EWS are observed, with highest numbers found for $s_1 = 10$ years and $s_2 < 60$ years (Fig. 4c). The highest numbers of simultaneous EWS in both $\hat{w}^2$ and $\hat{H}^{loc}$ is 7, found for $s_1 = 10$ years and $s_2 = 50$ or 60 years (Fig. 4e).

When, instead, searching for EWS in the entire interval between subsequent transitions, the averaged wavelet power $\hat{w}^2$ exhibits 13 significant EWS for $s_1 = 10$ years and $s_2 = 40$ years or $s_2 = 50$ years (Fig. 4b). However, $\hat{H}^{loc}$ exhibits substantially less significant EWS in this situation, with highest number equal to 6, observed for $(s_1, s_2) = (10, 20)$ years and $(s_1, s_2) = (70, 80)$ years (Fig. 4d). The highest numbers of simultaneous EWS in both estimators are 4 and 5, detected in the latter two periodicity bands (Fig. 4f).

## Discussion

The hypotheses that the DO transitions are caused by bifurcations in a dynamical subsystem operating at decadal timescales, which was first postulated by Rypdal[14], is thus confirmed by the results presented here, using the higher-resolution NGRIP $\delta^{18}O$ data.

The EWS detected here carry, despite the terminology, little predictive power: First, the above significance test on the number of EWS shows that a considerable number of false positives would be involved in a forecast based on these EWS. Second, a forecast issued on the basis of EWS would require to quantify a lead time, and it is unclear how to achieve this with observational data. Note that these issues are not specific to the results presented in this study but apply generally to EWS in empirical time series. To address them, substantial further research is needed; in particular, a very detailed reconstruction of the underlying attractor basin, alongside a precise estimation of the internal noise level, would be needed, and both would have to be put into quantitative relation with the changes in EWS estimators, such as variance or autocorrelation. Note that the restriction to specific frequency bands, and thus considering $\hat{w}^2$ and $\hat{H}^{loc}$ instead of the variance and autocorrelation, further complicates the problem of using the signals revealed here in a predictive setting.

The EWS presented here are, however, very useful from a different perspective, namely, in the context of relating statistical EWS with associated physical mechanisms. In the case at hand, the presence of EWS in both $\sigma^2$ and $\alpha_1$, as well as in the wavelet-based estimators $\hat{w}^2$ and $\hat{H}^{loc}$, obviously suggests that the DO events are neither directly externally forced nor purely induced by noise. Furthermore, the timescale at which the EWS are overall most pronounced, i.e. the 10–50 year periodicity band, helps to constrain the set of potential physical mechanisms responsible for the DO transitions: It is unlikely that oceanic circulation processes alone, such as changes in the Atlantic Meridional Overturning Circulation (AMOC[18,19]), would leave their imprint in such high frequencies. Hence, the covariability between NGRIP $\delta^{18}O$ and AMOC strength that is inferred from paleoclimate proxy data[20] is, on the basis of the results presented here, more likely to be due to a response of the AMOC to the bifurcation in the proposed, fast subsystem, than by a driving role of the AMOC. On the other hand, the dust concentration in ice cores is typically interpreted as a proxy for atmospheric circulation variability[21,22]. The fact that substantially less EWS are found in the dust time series therefore indicates that atmospheric processes are less likely to be the direct triggers of the DO events.

It has previously been suggested that the DO transitions in Greenland could be triggered by a rapid retreat of sea ice in the northern North Atlantic[23–27]. The $\delta^{18}O$ values in ice cores are not a direct proxy of the air temperature at the ice core site, because they depend, in fact, on the temperature gradient between the source region of the precipitation and the location of the ice core[28,29]. Indeed, a substantial cooling of the source temperatures around the DO events has been inferred from the deuterium excess[30]. There are, therefore, two mechanisms related to sea ice extent that affect this temperature gradient: First, a rapid retreat of sea ice, e.g. due to warming subsurface waters, would lead to the abrupt release of heat to the atmosphere, which was previously trapped below the ice[26,31]. This would enhance air temperatures at the ice core site and thereby reduce the temperature gradient to the source region. Second, rapid sea ice retreat would shift the evaporative source region of the water precipitating in Greenland northward and thereby reduce the temperature gradient as well. Both mechanisms would thus lead to an increase of $\delta^{18}O$ in the ice.

For these reasons, it is indeed very likely that variations of the extent of the sea ice cover are noticeable in the 10–50 year periodicity band of the NGRIP $\delta^{18}O$ time series, and the EWS signals reported here could hence be physically explained by enhanced fluctuations of the sea ice extent as this dynamical subsystem

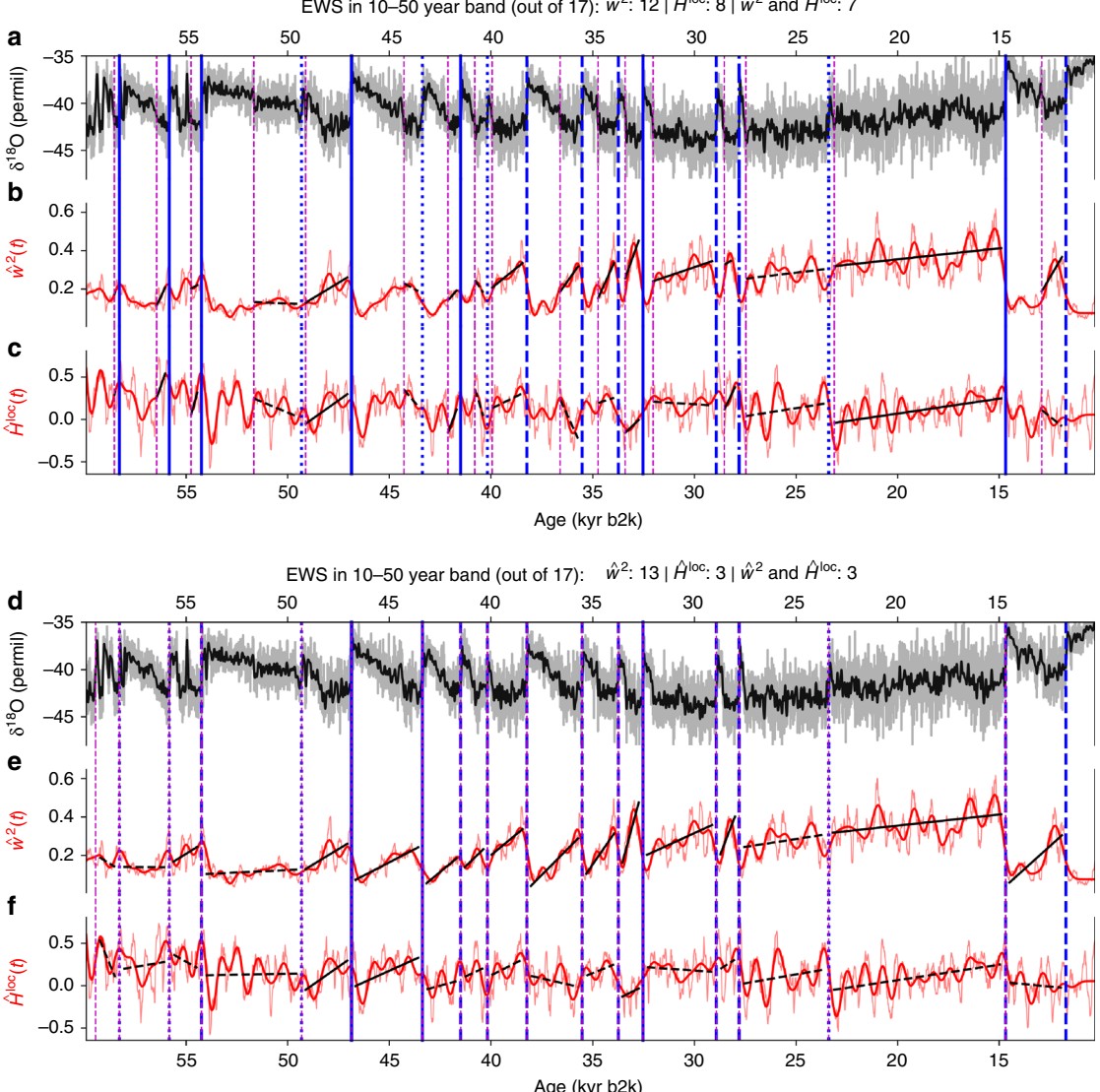

**Fig. 3** Early-warning signals in the wavelet-based estimators confined to the 10–50-year periodicity band. **a** The 5-year-interpolated $\delta^{18}O$ time series (grey), together with a low-pass filtered version (black) for visual clarity. Dansgaard–Oeschger (DO) events, as well as the Younger-Dryas/Preboreal transition, are marked by the vertical blue lines, and the preceding ends of GI intervals by vertical dashed magenta lines (cf. Supplementary Table 1 for corresponding ages). EWS are searched for in the GS intervals between the magenta and blue lines. **b** Time series of the wavelet-based estimator of high-frequency fluctuations $\hat{w}^2$, confined to the 10–50-year periodicity band (red, see text for details). **c** Same as **b** but for the local Hurst exponent $\hat{H}^{loc}$. **d–f** Same as **a–c**, but here, EWS are searched for in the entire intervals between subsequent DO transitions. Significant (non-significant) trends according to the phase-randomization test (see Methods) are indicated by the solid (dashed) black lines. Transition events that are preceded by significant EWS in $\hat{w}^2$ $\left(\hat{H}^{loc}\right)$ are marked by dashed (dash-dotted) blue lines, and transitions that are preceded by EWS in both $\hat{w}^2$ and $\hat{H}^{loc}$ are indicated by solid blue lines. If neither of the two estimators shows significant EWS, the blue lines are dotted

destabilizes on its way to the bifurcation. In this sense, the results presented here support hypotheses on mechanisms related to rapid sea ice retreat to explain the DO events.

## Methods

**Data preprocessing.** We employ the raw $\delta^{18}O$ and dust concentration data from the NGRIP at 5-cm depth resolution for the period from 60 to 11.7 kyears b2k[4,21,32]. In combination with the Greenland Ice Core Chronology 2005[16,17], which provides annual-layer-counted ages until approximately 60 kyears b2k, the $\delta^{18}O$ and dust values are interpolated to time series with regular 5-year sampling steps using cubic splines. The cumulative dating uncertainties, which arise from dating this record by counting annual layers, are not relevant for the kind of analysis presented in this study[33]. Figure 1 shows the temporal sampling steps between subsequent measurements and indicates that a temporal resolution of 5 years can be regarded as a lower bound[34], below which the possibility of biasing artefacts arising from the interpolation cannot be excluded: a fraction of 0.4% of the total number sampling steps is >5 years, but a comparably high fraction of 5.2%

is >4 years. Note that, for the dust record, we consider log(dust), because its values are approximately log-normally distributed[21].

**Scale-averaged wavelet coefficient and local Hurst exponent.** A comprehensive introduction to Wavelets can be found, e.g. in ref. [35]. For our analysis, we use the Paul wavelet basis (of order 4) because we are mainly interested in wavelet estimates that are sharp in time. However, corresponding results using the Morlet wavelet basis are also considered.

As an estimate of the wavelet power in a periodicity band between scales $s_1$ and $s_2$, we compute the following weighted average of the wavelet power spectrum $|W_t(s)|^2$, over all periodicities from $s_1$ to $s_2$:

$$\hat{w}^2(t) = \left\langle \frac{\delta j \delta t}{C_\delta} \sum_{j=j_1}^{j_2} \frac{|W_t(s_j)|^2}{s_j} \right\rangle_{200\,\text{years}}, \quad (1)$$

where $\langle \cdot \rangle_{\Delta t}$ denotes a moving average of width $\Delta t$, centred around $t$. For the reconstruction factor, we use $C_\delta = 1.132$ (ref. [35]), the resolution of frequency scales

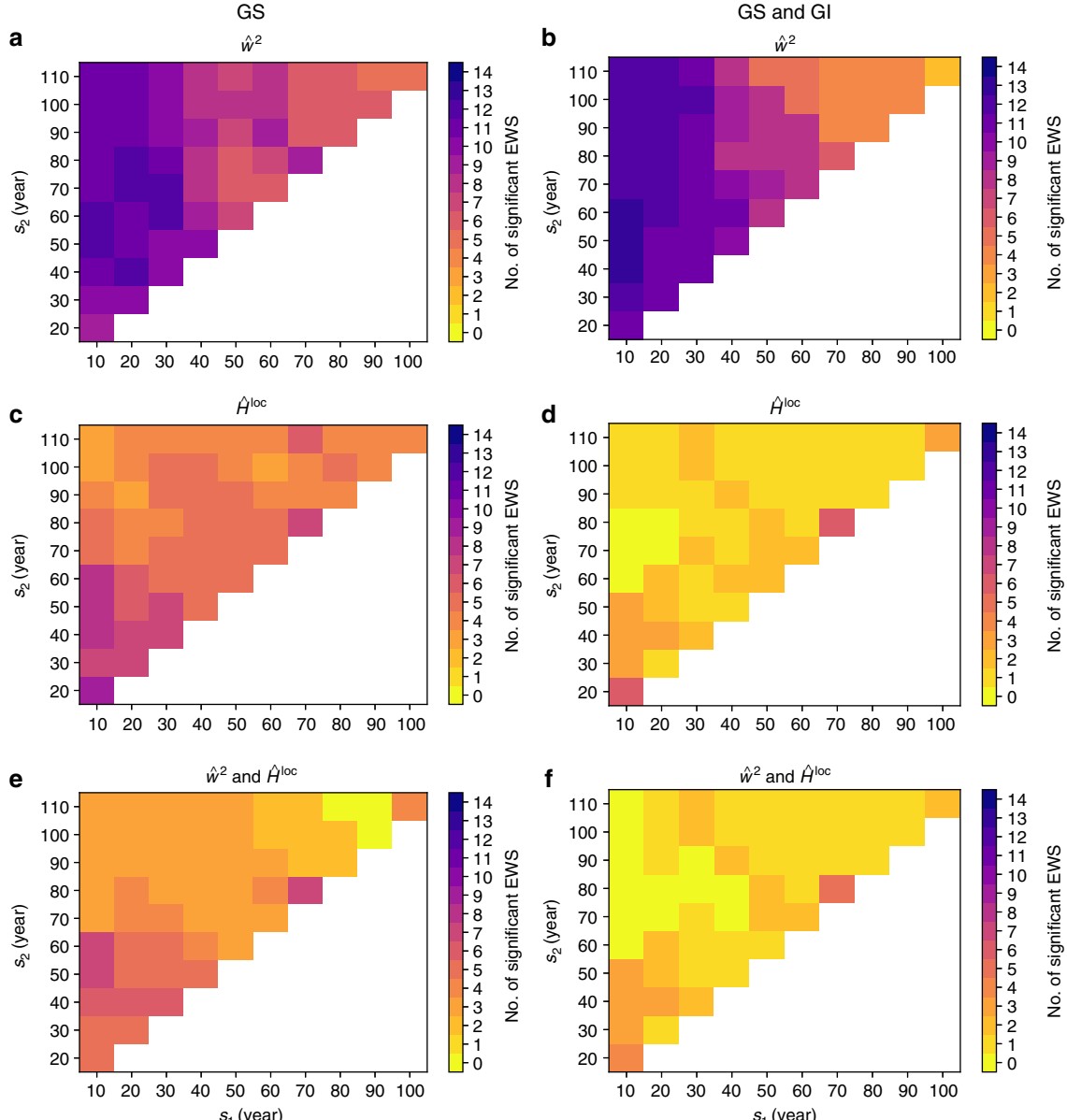

**Fig. 4** Numbers of significant wavelet-based EWS in different periodicity bands. Numbers of significant EWS for the $\delta^{18}O$ time series, as a function of $s_1$ and $s_2$, for EWS in $\hat{w}^2$ (**a**, **b**), in $\hat{H}^{loc}$ (**c**, **d**), and simultaneous EWS in both $\hat{w}^2$ and $\hat{H}^{loc}$ (**e**, **f**). The left column (**a**, **c**, **e**) shows the results for EWS in the GS intervals preceding transitions, while the right column (**b**, **d**, **f**) shows corresponding results when EWS are searched for in the entire period between subsequent transitions (GS and GI)

is set to $\delta j = 0.1$, and the temporal sampling resolution is $\delta t = 5$ years. Note that, in contrast to ref. [14], a weighted average of $|W_t(s_j)|^2$ is considered, in accordance with Eq. (24) in ref. [35]. This choice assures that, when summing over all scales, the total variance is recovered (see Eq. (14) in ref. [35]). Practically, the results on numbers of significant EWS do not depend on this choice.

For the local Hurst exponent $\hat{H}^{loc}(t)$, we have the scaling

$$\left\langle \frac{|W_t(s)|^2}{s} \right\rangle_{200\text{years}} \sim s^{2\hat{H}^{loc}(t)-1}, \qquad (2)$$

and $\hat{H}^{loc}(t)$ is thus, as in ref. [14], determined from the slope of a linear fit between $\log(s)$ and $\log\left\langle \left(|W_t(s)|^2/s\right)\right\rangle$, for $s_1 \le s \le s_2$. Finally, both $\hat{w}^2(t)$ and $\hat{H}^{loc}(t)$ are smoothed using a Chebyshev Type-I low-pass filter (with cutoff at 800 years) to extract the millennial-scale variability of these high-frequency components.

**Testing for significant trends**. The $\sigma^2$, $\alpha_1$, $\hat{w}^2$ and $\hat{H}^{loc}$ time series are, by construction, significantly autocorrelated. Therefore, the classical Mann–Kendal test for significance of trends is not applicable[36,37]. A null model based on phase-randomized surrogates preserves the spectral density and hence also the

autocorrelation structure and is thus more suitable. However, it should be noted that an existing trend leads to an overestimation of the autocorrelation[37,38]. Therefore, a linear model with autocorrelated noise is first fitted (using Maximum Likelihood) to each time series segment that is tested for EWS in terms of significantly positive trend in either of the estimators. If a non-zero trend is detected by this linear model, the trend is removed from the time series segment in question. Thereafter, surrogates are constructed from the de-trended segment by randomizing the phases in Fourier space[14,39,40]. For each time series segment, we construct 10,000 surrogates, compute the linear trend by fitting a linear model for each of them, and then derive a null model distribution for the linear slopes under the assumption that there is no trend (Supplementary Figure 4). Observed trends in $\sigma^2$, $\alpha_1$, $\hat{w}^2$ and $\hat{H}^{loc}$—correspondingly computed by fitting a linear model—are considered to be significant if their $p$ value with respect to these null model distributions is <0.05. Note that separate null models are constructed for each segment of $\sigma^2(t)$, $\alpha_1(t)$, $\hat{w}^2$ and $\hat{H}^{loc}$.

**Code availability**. All python code used for the analysis is available upon request via email (nboers@ic.ac.uk or boers@pik-potsdam.de).

**Data availability**. The raw NGRIP data are freely available at http://www.iceandclimate.nbi.ku.dk/data/.

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

## Acknowledgements
The author acknowledges funding by the Alexander von Humboldt Foundation, the German Federal Ministry for Education and Research, and the German Science Foundation. Helpful discussions with Michael Ghil and Andreas Groth are gratefully acknowledged.

## Author contributions
N.B. conceived the study, carried out the analysis, and wrote the paper.

## Additional information

**Competing interests:** The author declares no competing interests.

