## [Peer Review File · Nature Communications]

Reviewers' comments:

Reviewer #1 (Remarks to the Author):

This paper presents a new contribution to the ongoing discussion of precursors to the rapid stadial to interstadial transitions (Dansgaard-Oeschger events) observed most pronouncedly in Greenland ice core isotope records. Wavelet analysis of the high-resolution NGRIP d18O isotope records for the periods preceding the 17 transitions observed after 60 kyr BP (i.e. in well dated, layer counted part of the record). Two statistical quantities are considered (1) the integrated power in the s_1 - s_2 band (different values of the band pass are used, with best results for $s_1=10$ yr, $s_2=40$ yr). (2) The scaling exponent for the spectrum within the band pass, defined as a local Hurst exponent. Using a standard Mann-Kendall test it is observed if there are significant positive trends in these two quantities in the periods prior to the transitions. Indeed, for 11 out of 17 transitions this is the case. The author interprets this result to be indicative of the dynamics of the transitions to be a bifurcation. The science is sound and several statistical tests are performed to support the results. I can recommend publication after considering the points below.

My main concern is the description of the results as early-warning signals (EWS). I think this misleading, in the sense that if there is a signal increasing linearly in time prior to a transition and this signal is not crossing a pre-known threshold some time before the transition, there is no predictive power in the signal. That is different from the "standard" EWS: Increased variance and autocorrelation. These diverge at the bifurcation point (fluctuation-dissipation theorem and critical slow down), thus for those threshold-crossings can be defined and risks of false negatives and false positives can be estimated. I thus suggest either to speculate if this can be shown to be the case for these indicators as well. At least the local Hurst exponent might be related to critical slow down. Or, which I think is the better way to go, state that this is not about predicting (warning) when transition happens, it is about dynamical system identification from observed time series realization.

The author could make a stronger case if he could come up with an example of a bifurcation where the periodic variations (in the relevant frequency band) prior to the bifurcation should increase. Could this be a destabilization of a focus, a (transient) Hopf-bifurcation or something similar? I cannot see how there would be a periodic response prior to a saddle-node bifurcation.

It would be nice (and maybe even more informative than the Mann-Kendal significance plot) to see a histogram of the linear slopes in the two indicators (black lines on top of red curves in Fig 2). This would give a direct idea of how significant a (large) positive slope is in the GS. (Likewise a scatter plot of slopes for $\hat{\sigma}^2(t)$ and $\hat{H}^{\text{loc}}(t)$)

Minor points:

Line 225: induced noise -> noise induced

Line 236-238: That EWS are not found in dust implying that atmospheric processes can be ruled out is a very strong statement! Many researchers will oppose that!

Methods: Explanations could be expanded a little: why $\frac{d_j \Delta t}{C_{\Delta t}}$ an not just a single normalization constant (and what are the significances of these parameters?). Is the s_j in the denominator related to a $\log(s)$ weighting on frequencies? Please explain. Likewise why $W(s)^2/s \sim s^{2H-1}$ and not just $W(s)^2 \sim s^{2H}$?

Perhaps Fig 2 and Fig 3 could be merged to one figure. That would make it much easier to contrast the results for the two different frequency bands.

Reviewer #2 (Remarks to the Author):

Review of "Early-warning signals for Dansgaard-Oeschger events in a very high-resolution ice core record" by N. Boers.

The manuscript concerns the Dansgaard-Oeschger (DO) events, which are abrupt changes in climate that occurred repeatedly during the last ice age. Our knowledge about the events comes primarily from the Greenland ice cores, but they can also be detected in other climate reconstructions. From the ice core data, it is difficult to conclude on the cause of the events. But if one could observe indicators of stability weakening, it would constrain the set of plausible explanations for the DO events. Such a result would have great impact on our fundamental understanding of abrupt climate change, and it would convey a serious message about the risk of climate-system instability associated with anthropogenic influence.

The paper of Boer follows up on the hypothesis put forward by M. Rypdal, that the DO events are indeed linked to stability-weakening, but that the expected early-warning signals are masked by the low-frequency climate variability that results from the high thermal inertia of the deep ocean. By using the techniques proposed by Rypdal, he investigates the high-frequency band of the ice-core records on a new high-frequency version of the North Greenland Ice Core Project (NGRIP) data set. The results confirm Rypdal's hypothesis, and clearly establishes that rapid warming events are associated with reduced stability of dynamical processes operating on decadal time scales. The work represents a break-through in our understanding of the DO events.

General comments on the manuscript: The paper is well-written and easy to read. The following comments could help improve the presentation.

1. The word "very" in the title is not needed. What about replacing it with "new"?
2. Is it possible to state even more clearly that the paper resolves a scientific question with two competing hypotheses. It would highlight the importance of the work.
3. The sentences from line 61 to line 68 are hard to read. Could the author rewrite this part?

A comment on the statistical method: Boer uses a Mann-Kendall test for trend significance, but does not argue for how the problem of serial correlations is handled. This is a serious concern that needs to be addressed. A Monte Carlo method with randomization of phases is an option to consider. The problem is partly taken care of by the Monte Carlo method described in the paragraph starting on line 127, but this argument cannot be used for individual events. Therefore, the statement that 15 out of 17 events see significant early-warning signals, could be problematic.

Minor comments:

1. On line 35. That something is consistent with a Poisson distribution is a statement without meaning unless it is specified with respect to which test/measure/analysis. Please specify.
2. The "loc"-superscript on \hat{H} should not be in italic. It refers to the word "local".
3. On line 147: I would drop the superscript P on the ρ .
4. Would also use "and" instead of "&". For instance, in Fig. 4. Unless this is in accordance with journal standards.

5. In table 1, as well as in the tables in the supplementary, I would have removed some of the black grid lines. Perhaps one line at the top and one at the bottom?

My recommendation is that the paper can be accepted after a major revision. The crucial point is the validity of the statistical test for EWS significance for the individual events.

Niklas Boers

Point-by-point responses to the referees' comments

Response to Reviewer # 1:

This paper presents a new contribution to the ongoing discussion of precursors to the rapid stadial to interstadial transitions (Dansgaard-Oeschger events) observed most pronouncedly in Greenland ice core isotope records. Wavelet analysis of the high-resolution NGRIP $\delta^{18}O$ isotope records for the periods preceding the 17 transitions observed after 60 kyr BP (i.e. in well dated, layer counted part of the record). Two statistical quantities are considered (1) the integrated power in the s_1 - s_2 band (different values of the band pass are used, with best results for $s_1=10$ yr, $s_2=40$ yr). (2) The scaling exponent for the spectrum within the band pass, defined as a local Hurst exponent. Using a standard Mann-Kendall test it is observed if there are significant positive trends in these two quantities in the periods prior to the transitions. Indeed, for 11 out of 17 transitions this is the case. The author interprets this result to be indicative of the dynamics of the transitions to be a bifurcation. The science is sound and several statistical tests are performed to support the results. I can recommend publication after considering the points below.

I thank the reviewer for the overall positive evaluation of my manuscript. Please find responses to your helpful comments, and a version of the manuscript with highlighted changes, below.

My main concern is the description of the results as early-warning signals (EWS). I think this misleading, in the sense that if there is a signal increasing linearly in time prior to a transition and this signal is not crossing a pre-known threshold some time before the transition, there is no predictive power in the signal. That is different from the 'standard' EWS: Increased variance and autocorrelation. These diverge at the bifurcation point (fluctuation-dissipation theorem and critical slow down), thus for those threshold-crossings can be defined and risks of false negatives and false positives can be estimated. I thus suggest either to speculate if this can be shown to be the case for these indicators as well. At least the local Hurst exponent might be related to critical slow down. Or, which I think is the better way to go, state that this is not about predicting (warning) when transition happens, it is about dynamical system identification from observed time series

realization.

I entirely agree with the reviewer that for predictive purposes, the usefulness of the “EWS” detected here can be called into question; in the original version of the manuscript, there were already several sentences arguing that EWS are in general problematic for predictive purposes because of the issue of false positives, as well as the issue of providing lead times for a forecast based on EWS. Furthermore, even for “traditional” EWS, the value of the threshold mentioned by the reviewer is not a priori known: Although estimates like variance or autocorrelation diverge in simple model systems when approaching a co-dimension one bifurcation, this divergence does, of course, not happen in real-world observational time series. I have added another sentence in the discussion of the revised version, stating that this is even more problematic when restricting the analysis to specific frequency bands.

I would like to mention here, however, that the estimators $\hat{\sigma}^2(t)$ (which I renamed to $\hat{w}^2(t)$ in the revised manuscript) and $\hat{H}^{loc}(t)$ that I use are closely related to the variance and autocorrelation restricted to high-frequency spectral bands: The correlation between $\hat{w}^2(t)$ and the standard deviation, computed from a high-pass filtered (at 100yr) version of the NGRIP time series, is 0.97, while the correlation between $\hat{H}^{loc}(t)$ and the correspondingly computed autocorrelation is 0.71. I have added a figure (the new Fig. 2), showing that significant numbers of EWS are also found if the high-frequency variance and autocorrelation are used instead of the wavelet-derived estimates. The main reason for focussing on the latter estimates is that they allow for a much better scale separation, which enabled me to study the numbers of EWS in specific frequency bands (Fig. 4). I have also added several sentences in the main text to clarify the relation between classical variance and autocorrelation and the wavelet-based estimators.

The author could make a stronger case if he could come up with an example of a bifurcation where the periodic variations (in the relevant frequency band) prior to the bifurcation should increase. Could this be a destabilization of a focus, a (transient) Hopf-bifurcation or something similar? I cannot see how there would be a periodic response prior to a saddle-node bifurcation.

I think that there is a slight misunderstanding here: I am looking for EWS in predefined frequency bands of the NGRIP time series. In the case at hand, I focussed on different high-frequency (i.e., low-periodicity) bands, but this does not mean that there are actually periodic components in these bands.

The hypothesis that is tested in this way is that there exists a dynamical subsystem operating at decadal time scales, which exhibits a fold bifurcation (or any other co-dimension one bifurcation, which is the class for which the presence of EWS can be proven) prior to the DO events. In other words, the subsystem's response is not periodic, but the response is visible in specific low-periodicity bands, thereby indicating the specific time scales at which the subsystem operates.

It would be nice (and maybe even more informative than the Mann-Kendal significance plot) to see a histogram of the linear slopes in the two indicators (black lines on top of red curves in Fig 2). This would give a direct idea of how significant a (large) positive slope is in the GS. (Likewise a scatter plot of slopes for $\hat{\sigma}^2(t)$ and $\hat{H}^{loc}(t)$)

Following a comment by the other reviewer, I have replaced the Mann-Kendal test by a surrogate test based on randomizing the phases in Fourier space. The reason for this is that all EWS estimators which I consider have – by construction – non-zero autocorrelation. In such a case, the Mann-Kendall test's null hypotheses of no trend is rejected too easily, because the Mann-Kendall test assumes independence of the data points. Randomizing the phases preserves the spectrum and hence also the autocorrelation of the time series, leading to a more suitable and much more restrictive significance test. This has changed the numbers of significant EWS, but since the test on the total number of EWS (Fig.4 and Fig. S1) is adjusted accordingly, the overall results remain strongly significant. The problem I see with showing histograms of the different slopes is that for each variable and each GS, a separate test with its own null model distribution is constructed, and I would thus have to show at least 34 additional plots. In particular, a certain trend might be significant given the null model constructed for one GS, but it might not be significant with respect to the null model of another GS. I have added a plot showing the example of DO-16 for both the variance and the autocorrelation in the SI.

Minor points:

Line 225: induced noise → noise induced

Thanks, I have changes this.

Line 236-238: That EWS are not found in dust implying that atmospheric

processes can be ruled out is a very strong statement! Many researchers will oppose that!

I agree, and have toned down this statement.

Methods: Explanations could be expanded a little: why $\frac{\delta j \delta t}{C_\delta}$ an not just a single normalization constant (and what are the significances of these parameters?). Is the s_j in the denominator related to a $\log(s)$ weighting on frequencies? Please explain. Likewise why $W(s)^2/s \sim s^{2H-1}$ and not just $W(s)^2 \sim s^{2H}$?

For the definition of the scale-averaged wavelet power and related quantities, I have followed (Torrence and Como: A Practical Guide to Wavelet Analysis, BAMS 1999), which is a standard reference on wavelet analysis that has been cited almost ten thousand times. I have added additional information on the constants C_δ , δj , and δt . Of course, the fraction $\frac{\delta j \delta t}{C_\delta}$ is constant in time, and does therefore not affect the analyzed trends. The average of $|W_t(s_j)|^2$ is weighted by the scales s_j to assure consistency with the total variance of the time series, which is obtained by summing over the entire frequency domain (see Eq. (14) in the paper by Torrence and Como):

$$\sigma^2 = \frac{\delta j \delta t}{C_\delta} \sum_{t=0}^N \sum_{j=0}^J \frac{|W_t(s_j)|^2}{s_j}$$

Omitting the weights in the sum, however, does not affect the detected trends. I would, nevertheless, prefer to keep them in order to remain consistent with Torrence and Como (1999). I expanded on the explanations of these formulae in the revised manuscript.

Perhaps Fig 2 and Fig 3 could be merged to one figure. That would make it much easier to contrast the results for the two different frequency bands.

Thanks for this suggestion, in the new version of the manuscript, these figures are merged into one.

Response to Reviewer # 2:

Review of “Early-warning signals for Dansgaard-Oeschger events in a very high-resolution ice core record” by N. Boers.

The manuscript concerns the Dansgaard-Oeschger (DO) events, which are abrupt changes in climate that occurred repeatedly during the last ice age. Our knowledge about the events comes primarily from the Greenland ice cores, but they can also be detected in other climate reconstructions. From the ice core data, it is difficult to conclude on the cause of the events. But if one could observe indicators of stability weakening, it would constrain the set of plausible explanations for the DO events. Such a result would have great impact on our fundamental understanding of abrupt climate change, and it would convey a serious message about the risk of climate-system instability associated with anthropogenic influence.

The paper of Boer follows up on the hypothesis put forward by M. Rypdal, that the DO events are indeed linked to stability-weakening, but that the expected early-warning signals are masked by the low-frequency climate variability that results from the high thermal inertia of the deep ocean. By using the techniques proposed by Rypdal, he investigates the high-frequency band of the ice-core records on a new high-frequency version of the North Greenland Ice Core Project (NGRIP) data set. The results confirm Rypdal’s hypothesis, and clearly establishes that rapid warming events are associated with reduced stability of dynamical processes operating on decadal time scales. The work represents a break-through in our understanding of the DO events.

Thank you for the overall positive evaluation of my manuscript. To avoid confusion, please note that in the revised version of my manuscript, I have (following a comment by the other reviewer) included an analysis of ‘classical’ EWS, i.e. direct variance and autocorrelation increases, for comparison with the wavelet-based estimators. Since I refer to the variance by σ^2 , I have renamed the scale-averaged wavelet coefficient from $\hat{\sigma}^2$ to \hat{w}^2 .

General comments on the manuscript: The paper is well-written and easy to read. The following comments could help improve the presentation.

1. The word “very” in the title is not needed. What about replacing it with “new”?

I removed the “very” from the title, but did not replace it with “new”, as this would seem quite relative to me.

2. Is it possible to state even more clearly that the paper resolves a scientific question with two competing hypotheses. It would highlight the importance of the work.

Thanks for this suggestion, I have added sentences in the abstract, introduction and the discussion to make this even clearer.

3. The sentences from line 61 to line 68 are hard to read. Could the author rewrite this part?

I agree, and have rewritten these sentences; I hope that they are clearer now.

A comment on the statistical method: Boer uses a Mann-Kendall test for trend significance, but does not argue for how the problem of serial correlations is handled. This is a serious concern that needs to be addressed. A Monte Carlo method with randomization of phases is an option to consider. The problem is partly taken care of by the Monte Carlo method described in the paragraph starting on line 127, but this argument cannot be used for individual events. Therefore, the statement that 15 out of 17 events see significant early-warning signals, could be problematic.

I thank the reviewer for this comment. This is, indeed, a serious issue, which in fact applies to most studies analyzing EWS that I am aware of. I had not considered this point before, and apologize for this oversight on my part. In the revised version of the manuscript, I followed the reviewer’s suggestion: the significance of trends is not tested with Kendall’s tau anymore, but rather by determining linear trends directly, and comparing with a null model derived from randomized phases, in a similar way as in (Rypdal, Journal of Climate 2016). A subtle point in this context is, however, that an actually existing trend will lead to an overestimation of the autocorrelation (see Fig. 1 of this response letter, as well as [X. Zhang and F. Zwiers: *Comment on “Applicability of prewhitening to eliminate the influence of serial correlation on the Mann-Kendall test” by Sheng Yue and Chun Yuan Wang, Water Resources Research 2004*]). Therefore, a superposition of a linear model and an AR1 process is first fitted to each time series segment using Maximum Likelihood Estimation (as suggested also by Zhang and Zwiers),

and if a trend is present in this linear model, it is first removed from the time series segment before constructing the phase-randomized surrogates.

Figure 1: Influence of trends on the estimation of the AR-1 coefficient a . We construct ensembles of white noise surrogates and add increasing linear trends. We then test, for each value of the imposed trend separately, the null hypotheses that the AR-1 coefficient is equal to zero, with 95% confidence level. The corresponding significance threshold is determined beforehand, as the 95th percentile of the distribution of AR-1 coefficients estimated from white noise surrogates with no trend. The blue curve shows the rejection rate of this test, for different values of the imposed trend. For trend equal to zero, the rejection rate is 0.05, as expected from the confidence level. However, for increasing positive imposed trends, the rejection rate increases, because the trend leads to an overestimation of the AR-1 coefficient. The red line shows the corresponding estimate of the AR-1 coefficient, indicating that this strongly increases with increasing trend, although the time series are constructed from a linear model with white noise.

This revision has changed the number of significant EWS: The maximum number for $\hat{\sigma}^2$ is now 13 (instead of 15), and for \hat{H}^{loc} the maximum number is 8 (instead of 12). However, since the statistical test for the total number of significant EWS is adjusted accordingly, the obtained results are still highly significant. Therefore, adjusting the significance test does not affect the statement that a significant number of EWS is found, and any of the interpretations of the results in terms of a physical subsystem operating at scales below 50 yr to trigger the DO events.

Minor comments:

1. On line 35. That something is consistent with a Poisson distribution is a statement without meaning unless it is specified with respect to which test/measure/analysis. Please specify.

I agree, and have added that the distribution of waiting times between subsequent DO events is, with high likelihood, an exponential distribution; this implies that the null hypotheses of a Poisson process causing the DO transitions cannot be rejected. For details, please see the corresponding reference (Ditlevsen et al., *Climate of the Past* 2007).

2. The ‘loc’-superscript on \hat{H} should not be in italic. It refers to the word ‘local’.

Thanks, I have changed this throughout the manuscript.

3. On line 147: I would drop the superscript P on the ρ .

Thanks, I have followed this suggestion.

4. Would also use “an” instead of “&”. For instance, in Fig. 4. Unless this is in accordance with journal standards.

I have changed this the the table and figure legends.

5. In table 1, as well as in the tables in the supplementary, I would have removed some of the black grid lines. Perhaps one line at the top and one at the bottom?

I have not changed this because, if I’m not mistaken, the format of the tables will be changed to the Journal’s standards in case of acceptance.

My recommendation is that the paper can be accepted after a major revision. The crucial point is the validity of the statistical test for EWS significance for the individual events.

REVIEWERS' COMMENTS:

Reviewer #1 (Remarks to the Author):

The paper has now been revised taking both reviewers comments adequately into consideration. With that I now recommend publication.

A few minor revisions though:

L 219, define s_1 and s_2 (endpoints of the frequency band)

now σ_1 is used for lag-one autocorrelation, inconsistent with α_1 in figure 2. I prefer α_1 or even better a_1 .

Figures, especially Fig. 4, labels are too small (and titles too big).

Reviewer #2 (Remarks to the Author):

I find that the points raised in the previous round of review (those of both reviewers) have been satisfactorily addressed. The manuscript is improved and I support publication in Nature Communications.